# Design of biochemical pattern forming systems from minimal motifs

**Philipp Glock[1†], Fridtjof Brauns[2,3†], Jacob Halatek[2,3,4†], Erwin Frey[2,3]\*, Petra Schwille[1]\***

[1]Max-Planck-Institute of Biochemistry, Martinsried, Germany; [2]Arnold Sommerfeld Center for Theoretical Physics, Department of Physics, Ludwig-Maximilians-Universität München, München, Germany; [3]Center for NanoScience, Department of Physics, Ludwig-Maximilians-Universität München, München, Germany; [4]Biological Computation Group, Microsoft Research, Cambridge, United Kingdom

**Abstract** Although molecular self-organization and pattern formation are key features of life, only very few pattern-forming biochemical systems have been identified that can be reconstituted and studied in vitro under defined conditions. A systematic understanding of the underlying mechanisms is often hampered by multiple interactions, conformational flexibility and other complex features of the pattern forming proteins. Because of its compositional simplicity of only two proteins and a membrane, the MinDE system from *Escherichia coli* has in the past years been invaluable for deciphering the mechanisms of spatiotemporal self-organization in cells. Here, we explored the potential of reducing the complexity of this system even further, by identifying key functional motifs in the effector MinE that could be used to design pattern formation from scratch. In a combined approach of experiment and quantitative modeling, we show that starting from a minimal MinE-MinD interaction motif, pattern formation can be obtained by adding either dimerization or membrane-binding motifs. Moreover, we show that the pathways underlying pattern formation are recruitment-driven cytosolic cycling of MinE and recombination of membrane-bound MinE, and that these differ in their in vivo phenomenology.

**\*For correspondence:**
frey@lmu.de (EF);
schwille@biochem.mpg.de (PS)

[†]These authors contributed equally to this work

**Competing interests:** The authors declare that no competing interests exist.

## Introduction

Patterns are a defining characteristic of living beings and are found throughout all kingdoms of life. In the last years, it has become increasingly clear that protein patterns formed by reaction–diffusion mechanisms are responsible for a large range of spatiotemporal regulation (*Green and Sharpe, 2015*). Such processes allow organisms and cells to achieve robust intracellular patterning rooted in basic physical and chemical principles.

However, there is a lack of mechanistic understanding of the relationship between biomolecular features of proteins, that is their interaction domains and conformational states, and the collective properties of protein networks resulting in self-organized pattern formation. In other words, it is often unclear what exactly constitutes a *mechanism* of self-organization *on the biochemical level*. A major question is to what degree system-level biological functions, for example geometry sensing or length-scale selection, depend on particular biomolecular features. Some of these features may be essential for function, others may be irrelevant or redundant. The ability to unravel this *feature–function relationship* crucially depends on our ability to reconstitute biochemically distinct minimal systems experimentally and to compare these minimal variants to corresponding quantitative theoretical models. The key merit of such a combined approach is the ability to dissect different network architectures and also explore a broad range of reaction rates, and thereby uncover biomolecular mechanisms for system-level properties.

Here, we address this feature-function relationship in the context of a fairly well-understood biological pattern-forming system: the Min-protein system of *Escherichia coli*. All its components are known – only two proteins are needed to form the pattern (MinD and MinE) – and the system has been successfully reconstituted in an easily malleable in vitro system (*Loose et al., 2008*; *Ivanov and Mizuuchi, 2010*; *Vecchiarelli et al., 2014*; *Caspi and Dekker, 2016*; *Kretschmer et al., 2017*). In the bacterial cell, this system contributes to the positioning of FtsZ, a key component of the division ring, at mid-cell. Two proteins, MinD and MinE, oscillate between the cell poles and thereby form a concentration gradient with a minimum at mid-cell. MinC, piggybacking on MinD, consequently inhibits FtsZ polymerization at the poles and thus positions the Z-ring in the middle.

Even though the Min protein system seems simple at first glance, there is much (and biologically relevant) complexity within the protein domain sequences and structures, and hence in the interaction between proteins. MinD is an ATPase which is believed to dimerize upon ATP-binding, raising its membrane affinity via the C-terminal membrane targeting sequence (MTS) (*Lackner et al., 2003*; *Hu et al., 2002*; *Szeto et al., 2003*). Bound to the membrane, MinD recruits further MinD-ATP, as well as its ATPase-activating protein MinE, which together form membrane-bound MinDE complexes (*Hu and Lutkenhaus, 2001*; *Hu et al., 2002*). MinE stimulates MinD's ATPase activity, thereby initiating disintegration of MinDE complexes and subsequent release of MinE and ADP-bound MinD into the cytosol. MinE, although only 88 amino acids in length, is a biochemically complex protein. It is found as a dimer in two distinct conformations (*Pichoff et al., 1995*; *Park et al., 2011*): While diffusing in the cytoplasm, both the N-terminal MTS and the sequence directly interacting with MinD are buried within the protein. Upon sensing membrane-bound MinD, these features are released, which allows interaction with both the membrane and MinD (*Park et al., 2011*).

In summary, MinE exhibits four distinct functional features: activating MinD's ATPase, membrane binding, dimerization, and a switch between an open, active and a closed, inactive conformation. The roles of these distinct functional features of MinE for pattern formation have previously been studied and discussed in the literature (*Vecchiarelli et al., 2016*; *Kretschmer et al., 2017*; *Denk et al., 2018*). It has been shown that MinE's conformational switch is not essential for pattern formation, but conveys robustness to the Min system, as it allows pattern formation over a broad range of ratios between MinE and MinD concentrations (*Denk et al., 2018*). Furthermore, membrane binding of MinE was found to be non-essential for pattern formation (*Kretschmer et al., 2017*). These previous studies essentially retained the structure of MinE, predominantly mutating single residues.

Here, we chose a more radical strategy, in order to attempt a minimal design of fundamental modules towards protein pattern formation from the bottom-up. Specifically, we reduced MinE to its bare minimum function: binding to MinD, and thereby catalyzing MinD's ATPase activity. We then reintroduced additional features—membrane binding and dimerization—one by one in a modular fashion, to study their specific role in pattern formation. This approach allowed us to identify the essential biochemical modules of MinE and show that these facilitate two biochemically distinct mechanisms of pattern formation. We further analyzed these mechanisms in terms of reaction–diffusion models using theoretical analysis and numerical simulation. In particular, we show that the dimerization-driven mechanism is likely to be the dominant one for in in vivo pattern formation.

## Results and discussion

Full flexibility and control over all parameters was achieved by reconstituting purified Min proteins and peptides in an in vitro well setup consisting of a glass-supported lipid bilayer with a large, open reservoir chamber (see Materials and methods section for further details). To minimize the complexity of MinE in this reconstituted experimental system, we removed all sequences not in direct contact with MinD, keeping only 19 amino acids (13–31, further referred to as minimal MinE peptide) (*Figure 1*). In agreement with previous studies (*Loose et al., 2008*; *Glock et al., 2018a*), we observed that the native in vitro Min system, consisting of MinD and full-length MinE, forms traveling (spiral) waves (see *Figure 2a*) and (quasi-)stationary patterns. In contrast, we did not observe pattern formation for the reconstituted system containing the minimal MinE peptide in the nanomolar to low micromolar range (see *Figure 2b*), suggesting that it lacks essential molecular features for pattern formation. Instead, membrane binding of MinD was dominant even for high concentrations of up to

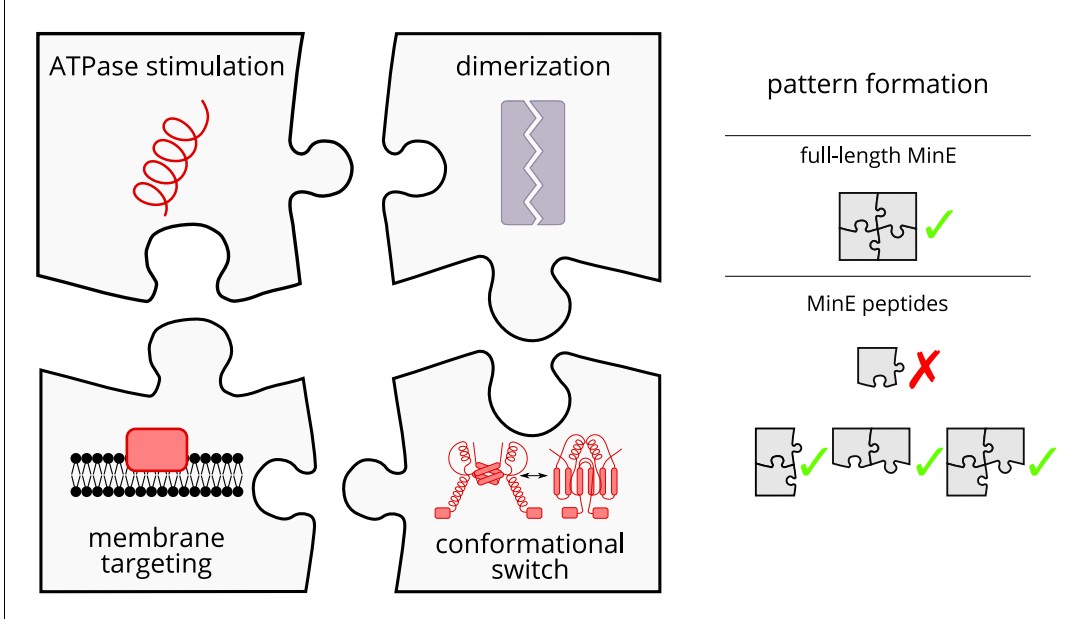

**Figure 1.** Schematic of the modular approach we took to engineering MinE in the in vitro Min system. While MinE has the core function to stimulate MinD's ATPase, three additional properties help MinE to facilitate the emergence of spatiotemporal patterns. We show that two of these properties, dimerization and membrane targeting, can be modularly added to a minimal MinE peptide to facilitate pattern formation.

20 μM of the minimal MinE peptide. We next tried to rescue pattern formation capability by re-introducing biomolecular features of MinE in a modular fashion.

Previous theoretical research has elucidated the key role of MinE cycling for the Min oscillations (*Halatek and Frey, 2012*). Each cycling step of MinE displaces one MinD from the membrane and thereby drives the oscillations that underlie pattern formation (*Halatek et al., 2018b*). Specifically, in this model, MinE is assumed to cycle between a cytosolic state and a MinD-bound state on the membrane. To facilitate pattern formation, this cytosolic-cycling mechanism requires sufficiently strong recruitment of cytosolic MinE by membrane-bound MinD (*Halatek and Frey, 2012*) suggesting that the recruitment rate of the minimal MinE peptide is too low. As the native MinE is a dimer, we hypothesized that dimerization might lead to increased recruitment, thus rescuing pattern formation. To test this hypothesis, we introduced dimerization back to the minimal MinE peptide by synthetically fusing it with well-described human and yeast leucine-zippers. Specifically, we cloned and expressed each construct with three different dimerization domains: Fos, Jun and GCN-4 (*Figure 1*) (*Szalóki et al., 2015*; *O'Shea et al., 1989*). Indeed, this modification enabled sustained pattern formation in the system (see *Figure 2d*). Compared to native MinDE patterns, those formed by dimerized peptides have larger wavelengths and are less coherent.

Another feature of native MinE that has been discussed in the context of pattern formation is persistent membrane binding via a membrane targeting sequence (MTS) (*Loose et al., 2011*). The MTS is located at positions 2–12 of the protein and allows MinE to remain membrane-bound after its interaction with MinD, that is it decreases the detachment rate of MinE. This persistent MinE-membrane binding facilitates that, after the dissociation of a MinDE complex, the freed-up MinE can bind to another MinD on the membrane, without cycling through the cytoplasm/bulk. Free, membrane-bound MinE is able to form a MinDE complex with membrane-bound MinD. As a shorthand, we will call this process *membrane recombination* of MinE. This process might alleviate the requirement for recruitment of MinD from the cytosol by membrane-bound MinD. To test whether the persistent membrane-binding of MinE can facilitate pattern formation, we added back the MTS found in native MinE (residues 2–12) to the N-terminus of the peptide. This construct, contrary to published results (*Vecchiarelli et al., 2016*), forms patterns with MinD. As shown in *Figure 2c*, the observed patterns are traveling waves with wavelengths several orders of magnitude larger than those found for the native in vitro Min system. Patterns are sustained over many hours within our assay.

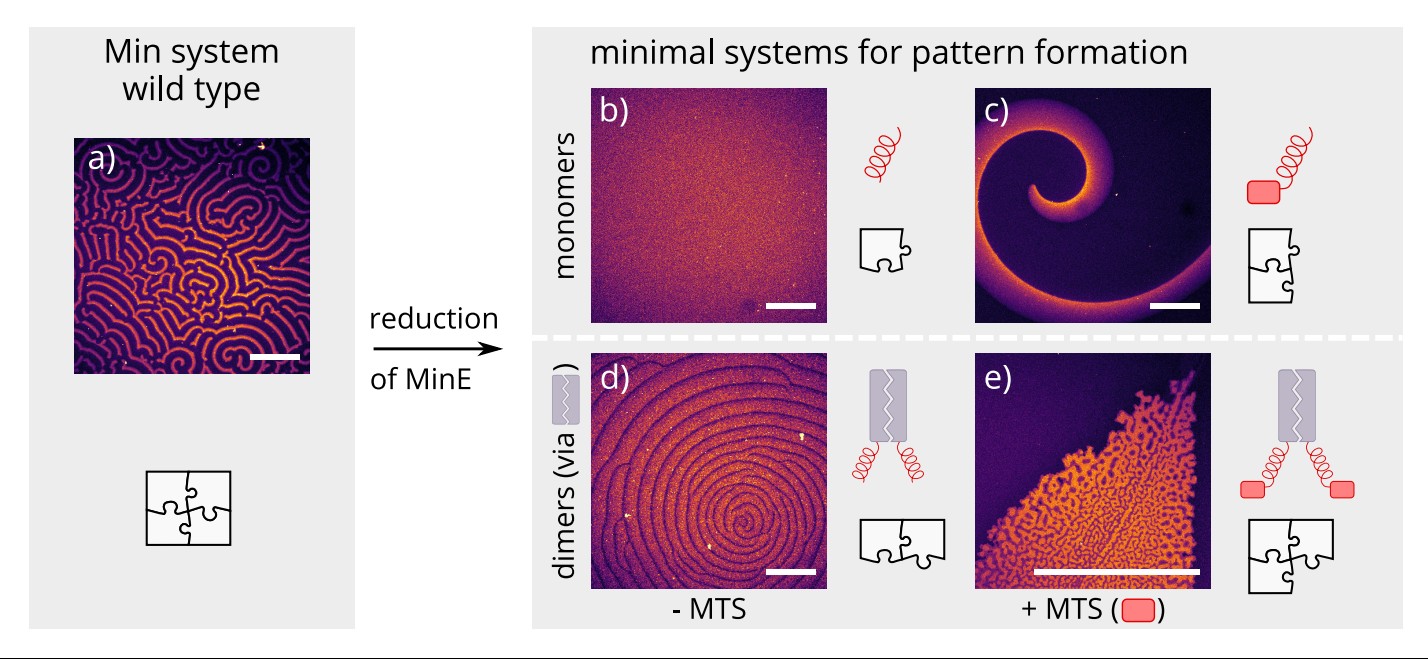

**Figure 2.** Patterns formed by the wild-type Min system and our minimal biochemical interaction networks. (**a**) MinD and MinE self-organize to form evenly spaced travelling waves when reconstituted on flat lipid bilayers. (**b**) The minimal MinE peptide capable of ATPase stimulation is MinE(13-31); it does not facilitate pattern formation. (**c**) The fragments MinE(1-31) and MinE(2-31)-sfGFP contain the membrane-targeting sequence (MTS) in addition to the ATPase stimulation domain. Substituting MinE with these constructs leads to pattern formation; see *Figure 2—video 1–3*. (**d**) Fusing the ATPase stimulation domain MinE(13-31) with dimerization domains (we tested Fos, Jun, or GCN-4) facilitates pattern formation in the absence of the MTS. (**e**) Combining membrane targeting and dimerization in a single construct produces quasi-stationary patterns. (Concentrations and proteins used: (**a**) 1 µM MinD, 6 µM MinE-His; (**b**) 1.2 µM MinD, 50 nM MinE(13-31); (**c**) 1.2 µM MinD, 50 nM MinE(1-31); scalebars = 300 µM; (**d**) 1 µM MinD, 100 nM MinE(13-31)-Fos; (**e**) 1.2 µM MinD, 100 nM MinE(1-31)-GCN4. In all assays, MinD is 70 % doped with 30 % Alexa647-KCK-MinD).

The online version of this article includes the following video and figure supplement(s) for figure 2:

**Figure supplement 1.** Global view of pattern formation by minimal systems.

**Figure supplement 2.** Titration results for MinE(1-31) and MinE(2-31)-sfGFP.

**Figure 2—video 1.** MinE(1-31) forms chaotic patterns with MinD.

https://elifesciences.org/articles/48646#fig2video1

**Figure 2—video 2.** MinE(2-31)-msfGFP forms chaotic patterns with MinD (1.8 µM MinD and 50 nM MinE(2-31)-msfGFP-His on 2:1 DOPC:DOPG).

https://elifesciences.org/articles/48646#fig2video2

**Figure 2—video 3.** Patterns with vastly different length and timescales coexist and continually transition into one another at certain concentrations of MinD and MinE(2-31)-msfGFP (0.6 µM MinD and 75 nM MinE(2-31)-msfGFP-His on 2:1 DOPC:DOPG).

https://elifesciences.org/articles/48646#fig2video3

Combining both features, that is adding both the MTS and a dimerization sequence to the minimal MinE peptide, resulted in (quasi-)stationary patterns, but the exact outcome depended heavily on the starting conditions of the assay (see *Figure 2e*). In general, patterns formed by MinD and our minimal MinE peptides do not show the same degree of order as patterns formed by the wild-type Min proteins (*Glock et al., 2018a*) or MinD and His-MinE (*Loose et al., 2008*). In particular, there is no well-controlled characteristic length scale (wavelength), and the defined spirals or stationary patterns observed in the wild-type Min system are sometimes replaced by chaotic centers as shown in *Figure 2d*. The chaotic behavior is especially pronounced at high MinD concentrations (in this case with a minimal MinE plus MTS and sfGFP or MinE(1-31), respectively) (*Figure 2—video 1* and *Figure 2—video 2*).

Our experimental results suggest that two distinct features of MinE, dimerization and membrane binding, independently facilitate pattern formation of our reconstituted Min system with engineered, minimal MinE peptides. To support these conclusions and gain further insight into the mechanisms underlying pattern formation, we performed a theoretical analysis using a reaction–diffusion model that captures all of the above biomolecular features. We extended the Min 'skeleton' model

introduced in *Huang et al. (2003)*; *Halatek and Frey (2012)* by MinE membrane binding, similar to the extension considered in *Denk et al. (2018)*. In this model, dimerization of MinE is effectively accounted for by an increased MinE recruitment rate. We performed linear stability analysis of the reaction–diffusion system to find the parameter regimes where patterns form spontaneously from a homogeneous initial state. The two-parameter phase diagram shown in *Figure 3a* shows that increased MinE recruitment as well as slower MinE detachment can rescue pattern formation, via two independent cycling pathways of MinE: cytosolic cycling and membrane recombination. This shows that our hypothesis that dimerization increases recruitment of MinE to MinD is consistent with the experimental findings.

To test whether either or both of these two pattern-forming pathways fulfill the biological function of the Min-protein patterns, we studied pattern formation using the generalized reaction–diffusion model taking into account realistic cell geometry. In *E. coli*, Min oscillations have to take place along the long axis of the rod-shaped cells for correct positioning FtsZ at midcell. Interestingly, linear stability analysis (see *Figure 3—figure supplement 4*) shows that the membrane-recombination-driven mechanism favors short-axis oscillations which is at odds with the biological function of the Min system. Indeed, our numerical simulations show that pole-to-pole oscillations are only possible for sufficiently strong cytosolic cycling, whereas the recombination-driven mechanism leads to side-to-side oscillations (see *Figure 3b*). A recent theoretical study on axis-selection of the PAR system in *Caenorhabditis elegans* suggests that pattern formation driven by an antagonism of membrane bound proteins generically leads to short-axis selection (*Gessele et al., 2018*). Here, membrane-bound MinE antagonizes membrane-bound MinD via the membrane-recombination pathway. Sufficiently strong MinE-recruitment from the cytosol supersedes the membrane-recombination pathway and leads to long-axis selection (pole-to-pole oscillations) even when MinE-membrane binding is strong.

Taken together, we conclude that Min-pattern formation in vivo is driven by cytosolic cycling of MinE, because correct axis selection (pole-to-pole oscillations) is essential for cell-division of *E. coli* and other gram-negative bacteria. In a broader context, our results demonstrate that multiple mechanisms with different characteristics, for example in their ability to sense geometry, can coexist in one reaction network. Most importantly, this highlights that a classification of pattern-forming mechanisms in terms of the reaction network topology alone misses important aspects of pattern formation that can be crucial for the biological function.

With respect to a potential biochemical origin of the pattern-forming mechanisms, we showed how additional protein domains can move the whole system into a mechanistically distinct regime. Enhancing the strength of MinE recruitment by MinD via dimerization shifts the system into a regime of recruitment-driven pattern formation. Alternatively, adding membrane targeting to the peptide unlocked a new pathway and led to sustained patterns via MinD-MinE recombination on the membrane (see supplementary discussion in Appendix 1 for further details).

In conclusion, the concept of modular engineering of pattern formation through distinct protein domains adds an entirely new dimension to the Min system, and establishes it further as a paradigmatic model for studying the mechanisms underlying self-organized pattern formation. Now, defined modules can be added, removed and interchanged. Interestingly, our experimental findings provide evidence that the distinct functional modules of MinE need not be provided by native parts of the proteins, but can be substituted with foreign sequences. Moreover, the part of MinE that interacts with MinD can be added as a small peptide tag of 19 amino acids to any host protein (as shown for superfolder-GFP + MTS, *Figure 2—figure supplement 2*), leading to a chimera protein that inherits key properties, such as membrane-interactions and protein-protein interactions, from the host protein. The modular domains provide an experimental platform to systematically modify the molecular interactions. Together with systematic theoretical studies, this is a powerful and versatile tool to study the general principles underlying biological pattern formation in multispecies, multicomponent reaction–diffusion systems.

## Materials and methods

Most experimental methods used in this publication were exhaustively described in text and video in a recent publication (*Ramm et al., 2018*). We therefore describe these techniques only in brief. This publication also includes a detailed and complete materials table for our assay.

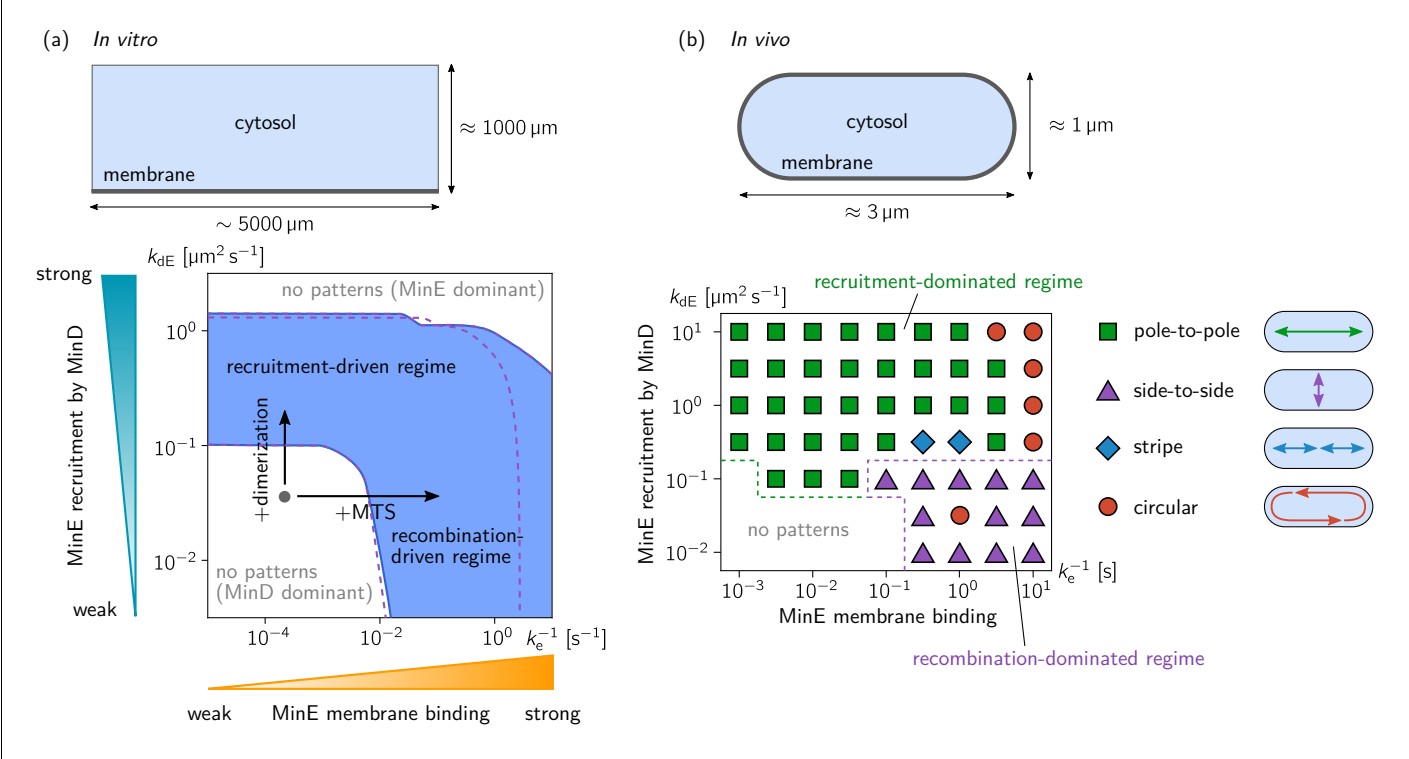

**Figure 3.** Pattern forming capability of the extended Min model in vitro and in vivo. (a) In vitro geometry and two-parameter phase diagram obtained by linear stability analysis, showing the pattern formation capabilities of the MinDE-system in dependence of MinE membrane-binding strength ($k_e^{-1}$) and MinE-recruitment rate $k_{dE}$. The regime of spontaneous pattern formation (lateral instability) is indicated in blue. The gray circle represents minimal MinE(13-31) construct, which does not facilitate self-organized pattern formation. The experimental domain additions are accounted for by respective changes of the kinetic rates, as indicated by the arrows. (Parameters: see Materials and methods; blue region: regime of pattern formation for zero MinE attachment, $k_E = 0$; purple dashed lines: boundary of the pattern-formation regime for non-zero MinE attachment rate, $k_E = 5$ μm s$^{-1}$). (b) Two-parameter phase diagram obtained by numerical simulations in in vivo geometry. We find regimes of different oscillation pattern types: pole-to-pole oscillations (green squares); side-to-side oscillations (purple triangles); stripe oscillations (blue diamonds); and circular waves (red circles). *Figure 3—videos 1–5* show examples each of these pattern types.

The online version of this article includes the following video and figure supplement(s) for figure 3:

**Figure supplement 1.** Network cartoon of the MinE 'skeleton' model extended by MinE membrane binding.

**Figure supplement 2.** Phase diagrams in the parameter plane of total concentrations ($n_E$, $n_D$).

**Figure supplement 3.** Phase diagrams showing how the range of MinE concentrations where the system is laterally unstable, depends on the MinE detachment rate and the MinE recruitment rate.

**Figure supplement 4.** Linear stability analysis in the ellipse geometry.

**Figure 3—video 1.** Pole-to-pole oscillation for weak MinE binding ($k_{dE} = 3.16$ μm$^2$s$^{-1}$, $k_e = 1000$ s$^{-1}$).

https://elifesciences.org/articles/48646#fig3video1

**Figure 3—video 2.** Pole-to-pole oscillation for strong MinE binding ($k_{dE} = 3.16$ μm$^2$s$^{-1}$, $k_e = 0.316$ s$^{-1}$).

https://elifesciences.org/articles/48646#fig3video2

**Figure 3—video 3.** Circular wave ($k_{dE} = 3.16$ μm$^2$s$^{-1}$, $k_e = 0.1$ s$^{-1}$).

https://elifesciences.org/articles/48646#fig3video3

**Figure 3—video 4.** Stripe oscillation ($k_{dE} = 0.316$ μm$^2$s$^{-1}$, $k_e = 3.16$ s$^{-1}$).

https://elifesciences.org/articles/48646#fig3video4

**Figure 3—video 5.** Side-to-side oscillation ($k_{dE} = 0.1$ μm$^2$s$^{-1}$, $k_e = 0.316$ s$^{-1}$).

https://elifesciences.org/articles/48646#fig3video5

## Membranes

SLBs were prepared from DOPC and DOPG (ratio 2:1) small unilamellar vesicles in Min buffer (25 mM Tris-HCl pH 7.5, 150 mM KCl, 5 mM MgCl$_2$) by adding them (at 0.53 mg/mL) on top of a charged, cleaned glass surface. The solution was diluted after one minute by addition of 150 mL Min buffer. After a total of 3 min, membranes in chambers were washed with 2 mL of Min buffer.

### Assay chamber

Assay chambers were assembled from piranha-cleaned coverslips and a cut 0.5 ml plastic reaction tube by gluing the tube upside down onto the cleaned and dried surface using UV-curable adhesive.

### In vitro self-organization assay

The buffer volume in an assay chamber containing an SLB was adjusted to yield a final volume of 200 μL including protein solutions and ATP. Proteins, peptides and further reactants were added and the solution was mixed by pipetting.

### Peptides

Peptides were synthesized using Fmoc chemistry by our in-house Biochemisty Core Facility. MinE(2-31)-KCK-Atto488 was expressed as a SUMO fusion in *E. coli* BL-21 DE3 pLysS cells, the SUMO tag was then cleaved using SenP2 protease and the remaining peptide was labelled using Atto488-maleimide to site-specifically target the cysteine residue. Labelling was done as described below.

### Protein design and purifications

Detailed information about cloning procedures and design of proteins can be found in the supplementary information.

### Protein concentration measurements

Protein concentrations were determined by using a modified, linearized version of the Bradford assay in 96-well format (*Ernst and Zor, 2010*).

### Labeling

Atto 488-maleimide in 5–7 μL DMSO (about three molecules of dye per protein) was added dropwise to ~0.5 mL of protein solution in storage buffer (50 mM HEPES pH 7.25, 300 mM KCl, 10 % glycerol, 0.1 mM EDTA, 0.4 mM TCEP) in a 1.5 mL reaction tube. The tube was wrapped in aluminium foil and incubated at 4° C on a rotating shaker for 2 to 3 hr. Free dye was separated from proteins first by running the solution on a PD-10 buffer exchange column equilibrated with storage buffer. Then, remaining dye was diluted out by dialysis against storage buffer overnight. The labeling efficiency was measured by recording an excitation spectrum of the labeled protein and measuring the protein concentration as described above. We then calculated the resulting labelling efficiency using the molar absorption provided by the dye supplier (Atto 488:$9.0 \times 10^4$ M$^{-1}$ cm$^{-1}$ ).

### Imaging

Microscopy was done on commercial Zeiss LSM 780 microscopes with 10x air objectives (Plan-Apochromat 10x/0.45 M27 and EC Plan-Neofluar 10x/0.30 M27). Tile scans with 25 tiles (5 × 5) at zoom level 0.6 were stitched to obtain overview images of entire assay chambers and resolve the large-scale patterns formed. More detailed images and videos were acquired on the same instruments using EC Plan-Neofluar 20x/0.50 M27 or Plan-Apochromat 40x/1.20 water-immersion objectives.

### The min 'skeleton model' extended by MinE membrane binding

To capture the effect of MinE membrane binding, we extend the 'skeleton' model introduced in *Halatek and Frey (2012)*. *Figure 3—figure supplement 1* shows a cartoon of the reaction network. We present the model first for a general geometry with a cytosolic volume coupled to a membrane surface. To perform linear stability analysis, we implemented this model in a 'box geometry' representing the in vitro setup with a membrane at the bottom, and in an ellipse geometry mimicking the rod-like cell shape of *E. coli*.

On the membrane, proteins diffuse and undergo chemical reactions, including attachment, detachment and interactions between membrane-bound proteins

$$\partial_t m_{\mathrm{d}} = D_m \, \nabla_m^2 m_{\mathrm{d}} + R_{\mathrm{d}}, \tag{1}$$

$$\partial_t m_{\mathrm{de}} = D_m \, \nabla_m^2 m_{\mathrm{de}} + R_{\mathrm{de}}, \tag{2}$$

$$\partial_t m_e = D_m \nabla_m^2 m_e + R_e, \tag{3}$$

where $\nabla_m$ is the gradient operator along the membrane. In the cytosol, proteins diffuse and MinD undergoes nucleotide exchange with a rate $\lambda$

$$\partial_t c_{DD} = D_D \nabla_c^2 c_{DD} - \lambda c_{DD} \tag{4}$$

$$\partial_t c_{DT} = D_D \nabla_c^2 c_{DT} + \lambda c_{DD} \tag{5}$$

$$\partial_t c_E = D_E \nabla_c^2 c_E \tag{6}$$

The two domains are coupled via the boundary conditions at the membrane

$$-D_D \nabla_\mathbf{n} c_{DD} = f_{DD}, \tag{7}$$

$$-D_D \nabla_\mathbf{n} c_{DT} = f_{DT}, \tag{8}$$

$$-D_E \nabla_\mathbf{n} c_E = f_E, \tag{9}$$

where $\nabla_\mathbf{n}$ is the gradient along the inward pointing normal (**n**) to the membrane. The reaction terms are derived from the interaction network *Figure 3—figure supplement 1* via the mass-action law and read

$$R_d = (k_D + k_{dD} m_d) C_{DT} - (K_{dE} c_E + k_{ed} m_e) m_d \tag{10}$$

$$R_{de} = (k_{dE} c_E + k_{ed} m_e) m_d - k_{de} m_{de}, \tag{11}$$

$$R_e = k_E c_E + k_{de} m_{de} - (k_e + k_{ed} m_d) m_e. \tag{12}$$

Correspondingly, the attachment-detachment flows are

$$f_{DT} = -(k_D + k_{dD} m_d) c_{DT}, \tag{13}$$

$$f_{DD} = k_{de} m_{de}, \tag{14}$$

$$f_E = k_{de} m_{de} - (k_E + k_{dE} m_d) c_E, \tag{15}$$

such that the dynamics conserve the global total densities of MinD and MinE

$$N_D = \int_{\text{mem}} dS (m_d + m_{de}) + \int_{\text{cyt}} dV (c_{DD} + c_{DT}), \tag{16}$$

$$N_E = \int_{\text{mem}} dS (m_e + m_{de}) + \int_{\text{cyt}} dV c_E. \tag{17}$$

## Linear stability analysis

To perform linear stability analysis, we need to find a set of orthogonal basis functions that fulfill the boundary conditions and diagonalize the Laplace operator, $\nabla^2$, on both domains (membrane and cytosol) simultaneously. In general, this is not analytically possible in arbitrary geometry. However, in a box geometry with a flat membrane, a closed form of the basis functions can easily be obtained. Furthermore, in a two-dimensional ellipse geometry, a perturbative ansatz can be used to obtain an approximate set of basis functions, as was shown in *Halatek and Frey (2012)* and used in *Wu et al. (2016)* and *Gessele et al. (2018)*. In the following, we briefly outline how the basis functions can be determined and employed to perform linear stability analysis. For details, we refer to the

supplementary materials of *Halatek and Frey (2018a)*, *Denk et al. (2018)*, *Halatek and Frey (2012)*, and *Gessele et al. (2018)*.

## In vitro box geometry

For linear stability analysis of the in vitro system, we consider a two-dimensional box with a membrane at the bottom surface, representing a slice through the in vitro system. The cytosol domain is a rectangle in the $x$–$z$ plane with height h and length L. The bottom boundary at $z = 0$ is the one-dimensional membrane domain – a line of length L. It is coupled to the bulk via reactive boundary conditions, *Equations (7) to (9)*. The other boundaries of the rectangular bulk domain are equipped with reflective boundaries. In this geometry, the gradient operators tangential and normal to the membrane are simply $\nabla_m \equiv \partial_x$ and $\nabla_{\mathbf{n}} \equiv \partial_z$.

The first step of a linear stability analysis is to calculate the steady state whose stability is to be analyzed. Typically this is a homogeneous steady state. In the system considered here, the most simple steady state is homogeneous along the x-direction. However, there must be cytosolic gradients in the z-direction due to the reactive boundary condition and the nucleotide exchange in the cytosol. Because the cytosol dynamics are linear, they can be solved in closed form.

To analyze the stability of such a steady state, one linearizes the dynamics around it. The ansatz to solve the resulting linear system is to diagonalize the Laplace operator. Importantly, in a system with multiple coupled domains, one needs to find a set of basis functions that diagonalize the Laplace operator on all domains (here membrane and cytosol), and that fulfill the reactive boundary conditions that couple these domains, simultaneously. In the x-direction, that is the lateral direction along the one-dimensional membrane, the eigenfunctions are simply Fourier modes. The bulk eigenfunctions in the z-direction, normal to the membrane, are exponential profiles and can be obtained in closed form by solving the linear cytosol dynamics, *Equations (4) to (6)*.

These eigenfunctions can then be plugged into the the membrane dynamics and the boundary conditions linearized around the homogeneous steady state. The resulting set of linear algebraic equations can be solved for the growth rates of the Fourier modes. Thus, one obtains a relationship between wavenumber q of a mode and its growth rate $\sigma(q)$. This relationship is called dispersion relation.

For details of the implementation of the linear stability analysis outlined above, we refer the reader to the supplementary materials of *Halatek and Frey (2018a)* and *Denk et al. (2018)*. Note that the bulk height dependence saturates above around 50 µm, the maximal penetration depth of bulk gradients (*Halatek and Frey, 2018a*). The bulk heights in the experiments were well above this saturation threshold at around 1 mm, allowing us to use the limit of large bulk height $h$.

## In vivo ellipse geometry

Linear stability analysis in an ellipse geometry is technically more involved, because the curved boundary makes it impossible to find a common eigenbasis of the Laplace operator on membrane and cytosol in closed form. For a detailed exposition of linear stability analysis in an elliptical geometry, we refer the reader to the supplementary materials of *Halatek and Frey (2012)*.

## Parameters

### In vitro

We used the kinetic rates and diffusion constants from *Halatek et al. (2018b)*; see *Table 1*. In this previous study, the Min skeleton model without MinE membrane binding was studied. Including MinE membrane binding leads to three additional kinetic rates in the model: We set the MinE membrane recombination rate to $k_{ed} = 0.1$ µms$^{-1}$, and varied the MinE detachment rate, $k_e$, in the range $10^{-1}$ µms$^{-1}$ to $10^5$ µms$^{-1}$. To test the effect of spontaneous MinE membrane attachment ($k_E > 0$) we compared the results from LSA for $k_E = 0$ and $k_E = 5$ µm s$^{-1}$, and found that spontaneous attachment is only relevant for very small MinE detachment rate, $k_e$, that is strong MinE membrane binding, where it suppresses pattern formation due to a dominance of membrane-bound MinE (see purple dashed line in *Figure 3a*).

For the $(k_e^{-1}, k_{dE})$ phase diagram (*Figure 3a*), the total densities of MinE and MinD were set to $n_E = 120$ µm$^{-2}$, $n_D = 1200$ µm$^{-2}$, corresponding to 0.1 µM MinE and 1 µM MinD in bulk solution, respectively. (Note that the unit for bulk concentrations is µm$^{-2}$ because we consider a two-

**Table 1.** Overview over the parameters used in the mathematical model.

In vitro parameters from *Halatek and Frey (2018a)*, in vivo parameters from *Halatek and Frey (2012)*; *Wu et al., 2016*. The diffusion constants, nucleotide exchange rate $\lambda$, and total protein densities are known from experiments *Loose et al. (2008)*; *Meacci et al. (2006)*. In *Halatek and Frey (2012)*, the kinetic rates of the Min skeleton model ($k_D$, $k_{dD}$, $k_{dE}$, and $k_{de}$) to reproduce the in vivo phenomenology quantitatively, and to optimize the biological function of the in vivo pole-to-pole oscillation (mid-cell localization). The additional rates ($k_{ed}$, $k_e$, and $k_E$) of the model extended by MinE-membrane binding are not constrained by experiment. We varied $k_e$ over several orders of magnitude (see *Figure 3* to study the role of persistent MinE-membrane binding. Note that, changing the MinE-recombination rate $k_{ed}$ over several orders of magnitude does not change our results qualitatively (topology of the phase diagrams).

| Name | Unit | In vitro | In vivo |
|---|---|---|---|
| $D_m$ | $\mu m^2\, s^{-1}$ | 0.013 | 0.013 |
| $D_D$ | $\mu m^2\, s^{-1}$ | 60 | 16 |
| $D_E$ | $\mu m^2\, s^{-1}$ | 60 | 10 |
| $\lambda$ | $s^{-1}$ | 6 | 6 |
| $n_D$ | $\mu m^{-2}$ | 1200 ($\approx$ 1$\mu$M) | $2000/V_{cell}$ |
| $n_E$ | $\mu m^{-2}$ | 120 ($\approx$ 1$\mu$M) | $700/V_{cell}$ |
| $k_D$ | $\mu m\, s^{-1}$ | 0.065 | 0.1 |
| $k_{dD}$ | $\mu m^2\, s^{-1}$ | 0.098 | 0.108 |
| $k_{dE}$ | $\mu m^2\, s^{-1}$ | 0.126 | 0.65 |
| $k_{de}$ | $s^{-1}$ | 0.34 | 0.4 |
| $k_{ed}$ | $\mu m\, s^{-1}$ | 0.1 | 0.2 |
| $k_e$ | $s^{-1}$ | $10^{-1}$ to $10^5$ | $10^{-1}$ to $10^3$ |
| $k_E$ | $\mu m\, s^{-1}$ | 0, 5 | 0, 5 |

dimensional slice through the three-dimensional bulk. The membrane concentrations have a unit $\mu m^{-1}$ respectively.)

In addition, we calculated $(n_E, n_D)$ phase diagrams at four points in $(k_e^{-1}, k_{dE})$ phase plane (see *Figure 3—figure supplement 2*). In these phase diagrams, one can see that mostly the E/D-concentration ratio, $n_E/n_D$, determines the regime of pattern formation. This is in *qualitative* agreement with the experimentally found phase diagram for the MinE(1-31) mutant (cf. *Figure 2—figure supplement 2*).

To exemplify how the critical E/D-ratio depends on the kinetic rates, we fixed the MinD concentration ($n_D$ = 1000 $\mu m^{-2}$) and varied $n_E$ and one of the kinetic rates. For the MinE-recombination driven regime, we set $k_{dE}$ = 0 (no MinE recruitment to MinD), and varied the MinE-detachment rate $k_e$ (see *Figure 3—figure supplement 3a*). The critical E/D-ratio of approximately 1/20 below which pattern formation is observed for the MinE(1-31) mutant in experiments is fitted for $k_e \approx$ 0.2 $s^{-1}$ (dashed red line and inset in *Figure 3—figure supplement 3a*). Note however, this 'fit' is severely underdetermined, because the remaining kinetic rates are not constrained by experiment. Changing, for instance, the MinE membrane recombination rate $k_{ed}$ (or any other kinetic rate) would lead to a different value for $k_e$ that fits the experimentally found concentration dependence. A remaining quantitative difference to the experimental findings is that the regime of pattern formation extends to very low MinE concentrations in the mathematical model, while there is a lower bound at a E/D-ratio of about 1/100 in the experiments.

*Figure 3—figure supplement 3b* shows the $(k_{dE}, n_E)$ phase diagram for the Min-skeleton model without persistent MinE-membrane binding (corresponding to $m_e \rightarrow \infty$).

## In vivo

We use the parameters from *Halatek and Frey (2012)* (see *Table 1*). In this previous study, the Min skeleton model was studied in vivo and the kinetic rates where fitted to reproduce the in vivo

phenomenology. The model extended by MinE membrane binding has three additional kinetic rates: We set the MinE membrane recombination rate to $k_{ed}$ = 0.2 µm s$^{-1}$, and varied the MinE detachment rate, $k_e$, in the range $10^{-1}$ s$^{-1}$ to $10^{-3}$ s$^{-1}$ . As in the in vitro case, spontaneous MinE membrane attachment ($k_E$>0) has no significant effect, so we set $k_E$ = 0. (Linear stability analysis and numerical simulations for a non-zero attachment rate $k_E$ = 5 µm s$^{-1}$ yield a phase diagram with the same qualitative structure as the one presented in *Figure 3b*.)

We mimic the cell geometry by an ellipse with lengths 0.5 m and 2 m for the short and long half axis, respectively (the corresponding cell 'volume' is $V_{cell}$ = 3.14 µm$^2$).

## Numerical simulations

The bulk-boundary coupled reaction–diffusion dynamics *Equations (1) to (15)* were solved using a finite element solver code (COMSOL Multiphysics).

Due to its large size, simulations of the in vitro system are very time consuming and beyond the scope of this work. Because most of the kinetic rates are not known, extensive parameter studies would be necessary to gain insight from such simulations.

## Acknowledgements

We thank the MPIB Biochemistry Core Facility, especially Stefan Pettera, for synthesizing peptides used in this study. We also thank Simon Kretschmer for stimulating discussions. PG acknowledges support by the International Max-Planck Research School for Molecular Life Sciences (IMPRS-LS). PG and FB acknowledge financial support by the DFG Research Training Group GRK2062 ('Molecular Principles of Synthetic Biology'). PS and EF acknowledge support by the DFG SFB1032 ('Nanoagents for the spatiotemporal control of molecular and cellular reactions', Project-ID 201269156).

## Additional information

### Funding

| Funder | Grant reference number | Author |
| --- | --- | --- |
| Deutsche Forschungsgemeinschaft | GRK2062 | Philipp Glock Fridtjof Brauns Erwin Frey Petra Schwille |
| Deutsche Forschungsgemeinschaft | SFB1032 (Project-ID 201269156) | Erwin Frey Petra Schwille |
| Max-Planck-Gesellschaft | International Max Planck Research School for Molecular and Cellular Life Sciences | Philipp Glock |

The funders had no role in study design, data collection and interpretation, or the decision to submit the work for publication.

### Author contributions

Philipp Glock, Conceptualization, Resources, Data curation, Validation, Investigation, Visualization, Methodology, Writing—original draft, Project administration; Fridtjof Brauns, Conceptualization, Resources, Data curation, Software, Formal analysis, Validation, Investigation, Visualization, Methodology, Writing—original draft, Project administration; Jacob Halatek, Conceptualization, Resources, Data curation, Software, Formal analysis, Supervision, Validation, Investigation, Methodology, Writing—review and editing; Erwin Frey, Petra Schwille, Conceptualization, Resources, Supervision, Funding acquisition, Methodology, Writing—review and editing

### Author ORCIDs

Philipp Glock  https://orcid.org/0000-0002-0238-2634
Fridtjof Brauns  https://orcid.org/0000-0002-6108-9278

Jacob Halatek (iD) https://orcid.org/0000-0003-3211-2253
Erwin Frey (iD) https://orcid.org/0000-0001-8792-3358
Petra Schwille (iD) https://orcid.org/0000-0002-6106-4847

**Decision letter and Author response**
Decision letter https://doi.org/10.7554/eLife.48646.sa1
Author response https://doi.org/10.7554/eLife.48646.sa2

## Additional files

**Supplementary files**
• Transparent reporting form

**Data availability**

All microscopy raw data, and Mathematica code simulation files (COMSOL Multiphysics) have been deposited in the Max Planck data service Edmond under the following URL: https://edmond.mpdl.mpg.de/imeji/collection/wGSlUmjVMnvxStN.

The following dataset was generated:

| Author(s) | Year | Dataset title | Dataset URL | Database and Identifier |
|---|---|---|---|---|
| Glock P, Brauns F, Halatek J, Frey E, Schwille P | 2019 | Design of biochemical pattern forming systems from minimal motifs - microscopy and additional data | https://edmond.mpdl.mpg.de/imeji/collection/wGSlUmjVMnvxStN | Edmond, wGSlUmjVMnvxStN |

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

**Appendix 1**

## Protein design and cloning

Several instances of MinE(2-31)-sfGFP were cloned, expressed and tested. We started with a construct carrying a His-tag on the N-terminus His-(MinE-2-31)-sfGFP. Then, we became concerned about dimerization of the fluorescent protein and introduced a mutation (V206K) to make His-MinE(2-31)-msfGFP. Then, we discovered that N-terminal tagging influences the properties of our minimal constructs and wt MinE and changed the construct to carrying a C-terminal His-tag (MinE(1-31)-msfGFP-His). The methionine residue was re-introduced here as a start codon, and is cleaved in *E. coli*. Additionally, we prepared MinE(13-31)-sfGFP and confirmed that without MTS, no patterns are formed.

The first construct, His-MinE(2-31)-sfGFP was cloned as follows: A fragment containing the pET28a vector-backbone and the start of His-MinE was amplified from pET28a-His-MinE using primers PG073+PG074. The sfGFP fragment was amplified from pVRB18-XX-sfGFP using primers PG069+PG070. The two fragments were recombined in *E. coli* to yield pET28a-His-MinE(2-31)-sfGFP. His-MinE(13-31)-sfGFP was assembled from three fragments. The sfGFP fragment was generated as described above. A second fragment containing the vector backbone and compatible overhangs was generated from pET28a-His-MinE using primers PG073+PG077. Finally, the MinE(13-31) fragment was amplified from pET28a-His-MinE using primers PG072+PG016, then a second PCR reaction was run on this fragment with primers PG076+PG074. All three fragments were recombined in *E. coli*.

His-MinE(2-31)-msfGFP was generated from His-MinE(2-31)-sfGFP by recombining two fragments generated by PCR with primers PG087+PG043 and PG088+PG044, respectively.

MinE(1-31)-msfGFP-His was recombined from two fragments. The MinE(1-31)-msfGFP was amplified from pET28a-His-MinE(2-31)-msfGFP using primers PG090+PG091. The vector fragment was generated from pET28a-BsMTS-mCherry-His using primers PG089+PG007.

Custom DNA sequences were ordered for GCN-4, c-Jun and c-Fos. DNA fragments consisting of a linker sequence, the respective leucine zipper and another linker sequence were amplified via PCR using primers PG103+PG104 (GCN4), PG105+PG106 (Jun) or PG107+PG108 (Fos). Similarly, FKBP and FRB were amplified using primers PG110+PG111 (FKBP) and PG112+PG113 (FRB). A fragment of MinE(13-31) containing compatible overlaps was generated from PCR on pET28a-MinEL-msfGFP-His using primers PG109+PG102. The vector containing MinE(1-31) and compatible overhangs was amplified from pET28a-MinE-His using primers PG007 and PG102. For the three-fragment assemblies, the vector was created via PCR from BsMTS-mCherry-His (Ramm, et al.) using primers PG007+PG089. The desired construct vectors were then created via three-fragment homologous recombination in *E. coli* TOP10, or two-fragment in case of MinE(1-31) constructs. In an additional step, the protein sequence KCK was inserted into the MinE(13-31) constructs by amplifying two halves of the vector. The first half was amplified using primers PG114+PG43, the second half using primers PG115+PG44. After DpnI digest (done for all fragments amplified from functional vectors), the fragments were transformed in to E. coli TOP10 and selected on kanamycin LB plates for homologous recombination. All constructs' integrity was verified via Sanger sequencing.

SUMO-MinE(1-31)-KCK-His and SUMO-MinE(13-31)-KCK-His were generated via homologous recombination of two fragments each. For the construct with MTS, one fragment was amplified from pET28M-SUMO1-GFP using primers PG043+PG116. The second fragment was amplified from pET28M-SUMO1-MinE (*Glock et al., 2018b*) using primers PG044+PG117. Fragments for the construct without MTS were amplified from the recombined vector described above using primers PG043+PG118 and from pET28M-SUMO1-GFP using primers PG044+PG119.

## Purification of proteins

MinD, MinD-KCK-Alexa647, mRuby3-MinD, His-MinE and MinE-His were purified as previously described (*Ramm et al., 2018*; *Glock et al., 2018a*; *Glock et al., 2018b*). MinE(13-31)-Fos, MinE(13-31)-Jun and MinE(13-31)-GCN4 were purified as described for MinE-His (*Glock et al.,*

*2018b*). MinE(2-31)-Fos, MinE(2-31)-Jun and MinE(2-31)-GCN4 were highly insoluble and therefore entirely found in the pellet fraction after cell lysis and centrifugation. The supernatant was discarded and the pellet re-solubilised in lysis buffer U (8M Urea, 500 mM NaCl, 50 mM Tris-HCl pH 8) by pipetting, vortexing and submerging the vial in a sonicator bath. The residual insoluble fraction was pelleted by centrifugation at 50,000 g for 40 min. The supernatant was incubated with Ni-NTA agarose beads (~2 mL per 400 mL initial culture) for 1 hr at room temperature on a rotating shaker. Agarose beads were pelleted at 400 g, 4 min and the supernatant was discarded. Purification was continued at RT since proteins were unfolded and kept in 8 M Urea. Agarose beads were loaded on a glass column and washed three times with 10 mL of above lysis buffer U. Further washes (3x) were performed with wash buffer U (8 M Urea, 500 mM NaCl, 20 mM imidazole, 50 mM Tris-HCl pH 8). The protein was eluted with elution buffer U (8 M Urea, 500 mM NaCl, 300 mM imidazole, 50 mM Tris-HCl pH8) and fractions with the highest protein content (Bradford, by eye) were pooled. Re-folding of the pooled eluate was done by dialyzing in multiple steps. In a first step, the solution was dialyzed against buffer D1 (6 M Urea, 500 mM NaCl, 50 mM Tris-HCl pH 8, 10% glycerol) over night. In a second step, against buffer D2 (4 M Urea, 500 mM NaCl, 50 mM Tris-HCl pH8, 10% glycerol) for 2 h, then against buffer D3 (2 M Urea, 500 mM NaCl, 50 mM Tris-HCl pH8, 10% glycerol) for further 2 h. The final dialysis was done against storage buffer (300 mM KCl, 50 mM HEPES pH 7.25, 10% glycerol, 1 mM TCEP, 0.1 mM EDTA). To separate the re-folded protein from aggregates, the protein solution was ultracentrifuged for 40 min at 50,000 g, 4°C. Protein concentration was then determined as described in the Materials and methods section. MinE(13-31)-KCK-His-Atto 488 and MinE(2-31)-KCK-His-Atto 488 were expressed and purified as described for MinE-His. SUMO-peptide fusions were then added into 1:100 (protease:protein) of SenP2 protease and dialyzed against storage buffer. Labeling was performed as described in the Materials and methods section.

## Protein concentration for dimerized constructs

For the dimerized constructs, not enough concentrations combinations were titrated to obtain full phase diagrams. We state here which combinations of protein concentrations we tested. All of the following conditions yielded pattern formation: 0.6 μM MinD, 50 nM MinE(13-31)-GCN4; 0.6 μM MinD, 50 nM MinE(13-31)-Jun; 0.6 μM MinD, 25 nM MinE(13-31)-Jun; 0.6 μM MinD, 30 nM MinE(13-31)-Jun; 0.5 μM MinD, 20 nM MinE(13-31)-Jun;0.5 μM MinD, 50 nM MinE(13-31)-Fos; 1 μM MinD, 100 nM MinE(13-31)-Fos; 1 μM MinD, 150 nM MinE(13-31)-Fos.

## Supplementary discussion

Going forward, it will be interesting to explore the Min system further along the avenue of individual protein domains/features and their role for self-organized pattern formation. We suspect that the minimization of MinE peptides could be taken even further by shortening the peptide. Especially at the C-terminus, we expect that several residues do not contribute to function, since they are not visible in a crystal structure of MinE(13-31) with MinD (*Park et al., 2011*). Additionally, the peptide still retains residues required for the dual function in the context of the MinE switch. Therefore, an optimized and further reduced peptide could be screened for. Additionally, our experiments with minimal peptides added to a superfolder–GFP (*Figure 2—figure supplement 2*) show that unrelated proteins can be attached. This opens the possibility to couple the spatiotemporal pattern to a different protein system. In principle, any protein can act as a minimal MinE if a peptide can be added internally or at either terminus of the protein.

Although we have not tested this prediction, we expect that the native MTS of MinE could be replaced with another MTS in our minimal peptides to restore pattern formation. It would be interesting to exchange the native MTS for a quantitatively described, diverse set of MTS to determine the required strength of membrane anchors needed for minimal MinE pattern formation. However, no such set or even just quantitative data on binding strength of multiple MTS is available at the moment.

Since we relate the lack of pattern formation to the recruitment rate of MinE(13-31), it may be possible to alter MinE recruitment by changing the buffer conditions such as salt concentration, type of ions (e.g. Sodium instead of Potassium), viscosity or pH. We can only speculate here, however, since screening a vast amount of conditions was not in the scope of the present study. Studies done on the wild type Min system using different buffer conditions showed some impact on pattern formation (*Vecchiarelli et al., 2014*; *Downing et al., 2009*).

## Primers used in this study

| Name | Sequence |
| --- | --- |
| PG007: AC-pET_for | GTCGAGCACCACCACCA |
| PG016: B-pET-MinE_r-ev | GTGCGGCCGCAAGCTTTTAGCGACGGCGTTCAGCAA |
| PG043: mut_KanR_fw | TGAAACATGGCAAAGGTAGCGT |
| PG044: mut_KanR_rev | GCTACCTTTGCCATGTTTCAGAAA |
| PG073: sfGFP-pET_fw | CATGGATGAGCTCTACAAATAAAAGCTTGCGGCCGCAC |
| PG074: sfGFP-li-MinE31_rev | AAAGTTCTTCTCCTTTGCTCACAGAACCAGAAGAACCAGAAGAGC-GACGGCGTTCAGCAAC |
| PG075: sfGFP-MinE_L_fw | CATGGATGAGCTCTACAAAGCATTACTCGATTTCTTTCTCTCGC |
| PG076: E-pET-MinEs_fw | GGGTCGCGGATCCGAATTCAAAAACACAGCCAACATTGCAA |
| PG077: lolipET_rv | GAATTCGGATCCGCGACC |
| PG087: sfGFP_V206K_fw | TACCTGTCGACACAATCTAAGCTTTCGAAAGATCCCAAC |
| PG088: sfGFP_V206K_rev | GTTGGGATCTTTCGAAAGCTTAGATTGTGTCGACAGGTA |
| PG089: pET28a-start_rev | CATGGTATATCTCCTTCTTAAAGTTAAACAA |
| PG090: pET-MinEL_fw | TAAGAAGGAGATATACCATGGCATTACTCGATTTCTTTCTCTCGC |
| PG091: pET-msfGFP_rev | TGGTGGTGGTGGTGCTCGACTCCAGATCCACCTTTGTAGAGCT |
| PG103: GCN4_fw | TCTTCTGGTTCTTCTGGTTCTCGTATGAAACAGCTGGAAGACAA |
| PG104: GCN4_rev | GTGCTCGACTCCAGATCCACCACGTTCACCAACCAGTTTTTTC |
| PG105: Jun_fw | TCTTCTGGTTCTTCTGGTTCTCGTATCGCTCGTCTGGAAGA |
| PG106: Jun_rev | GTGCTCGACTCCAGATCCACCGTAGTTCATAACTTTCTGTTTCAGCTG |
| PG107: Fos_fw | TCTTCTGGTTCTTCTGGTTCTCTGACCGACACCCTGCAG |
| PG108: Fos_rev | GTGCTCGACTCCAGATCCACCGTAAGCAGCCAGGATGAATTCC |
| PG109: pET-MinEs_fw | TAAGAAGGAGATATACCATGAAAAACACAGCCAACATTGCAAAAG |
| PG110: FKBP_fw | TCTTCTGGTTCTTCTGGTTCTGGTGTTCAGGTCGAAACTATCTCTC |
| PG111: FKBP_rev | GTGCTCGACTCCAGATCCACCTTCCAGTTTCAGCAGTTCAACG |
| PG112: FRB_fw | TCTTCTGGTTCTTCTGGTTCTGAAATGTGGCATGAGGGTCTC |
| PG113: FRB_rev | GTGCTCGACTCCAGATCCACCCTGTTTAGAGATGCGACGAAAGAC |
| PG114: li-KCK_fw | GGATCTGGAGTCGAGAAATGCAAACACCACCACCACCAC |
| PG115: li-KCK_rev | GTGGTGGTGGTGGTGTTTGCATTTCTCGACTCCAGATCC |
| PG116: KCK-pET_fw | GTTCTTCTGGTAAATGCAAATGAAAGCTTGCGGCCG |
| PG117: EL_li_rev | TTTGCATTTACCAGAAGAACCAGAACCGCGACGGCGTTCAGC |
| PG118: SUMO-Es-fw | ACCAGGAACAAACCGGTGGATCAAAAAACACAGCCAACATTGCAAA |
| PG119: SUMO_rev | TCCACCGGTTTGTTCCTGG |

