## [Decision Letter]

**Acceptance summary:**

This paper reports an interesting modular approach to dissect the molecular feature of the minD-minE biochemical network that allows for the generation of dynamical patterns of membrane composition. Four features of MinE are relevant to pattern formation; activating MinD's ATPase activity, membrane binding, dimerization, and a switch between an active and an inactive conformation. It was known from previous studies that structural switch and membrane binding are dispensable, while ATPase activation is required. This study shows that the ATPase activity is indeed essential, but must be coupled either membrane targeting or dimerization to generate patterns. The in vitro experimental study is nicely complemented with directly relevant modelling, and the modelling makes interesting predictions for how the variants would behave in the confined geometry of real *E. coli*.

**Decision letter after peer review:**

Thank you for submitting your article "Design of biochemical pattern forming systems from minimal motifs" for consideration by *eLife*. Your article has been reviewed by two peer reviewers, and the evaluation has been overseen by a Reviewing Editor and Naama Barkai as the Senior Editor. The reviewers have opted to remain anonymous.

The reviewers have discussed the reviews with one another and the Reviewing Editor has drafted this decision to help you prepare a revised submission.

Summary:

The minD-minE system is a very well study example of a biochemical network that generates dynamical patterns. The goal of this paper is to dissect which features among four – ATPase activity, membrane binding, dimerisation and conformation switch – of the minE protein is necessary for the emergence of patterns. It is known from previous studies that structural switch and membrane binding are dispensable, while ATPase activation is required. The authors show that the ATPase activity is indeed essential, but must be coupled either membrane targeting or dimerization to generate patterns. However, membrane targeting produces patterns of much larger wave-length then wild type, and dimerization lead to less coherent patterns (also with larger wavelength). Conjunction of the three features lead to patterns that are still less robust than wild type. These finding are rationalised within a pre-existing theoretical model via a modification of different reaction rates. The model is then use to predict how patterns produced by the different constructs in an in-vivo geometry.

The reviewers found this study novel and likely to be of general interest. They were particularly positive regarding the modular approach used to dissect the molecular requirement for patterning. The in vitro experimental study is nicely complemented with directly relevant modelling, and the modelling makes interesting predictions for how the variants would behave in the confined geometry of real *E. coli*.

However, there are several aspects that should be strengthened or clarified.

Essential revisions:

On the experimental side.

1) The reporting of experimental results rather succinct. One would in particular expect a more thorough analysis of the role of concentration. Figure 2—figure supplement 2. show only two titration curves, while three constructs lead to patterns. To what extent does the wave length of the pattern depend on concentration. These are important issues, since they can also be addressed by the model, and hence reinforce the comparison between the two approaches.

2) When studying the minimal, ATPase activating peptide of MinE, it is found that membrane binding by MinD remains dominant. It is surprising that at high peptide to MinD one does not see a lack of membrane binding by MinD as the peptide has been shown to stimulate the ATPase activity of MinD (paper by Goto in PNAS).

3) By adding a dimerization or MTS motif the authors find they can recover pattern formation (albeit different from each other). Does the dimerized peptide result in less coherent pattern than the I24N mutant used in a previous study? If so can the authors guess as to what feature of dimerization leads to this difference?

4) The text states "but the exact outcome depended heavily on the starting conditions of the assay". What exactly is meant by "starting conditions"? Does this refer to different concentrations, or do the authors observe qualitatively different outcomes even for the same nominal initial conditions including concentrations?

5) Given that minE dimerisation is sufficient to produce pattern of similar wavelength than the wild type, and that dimerisation with membrane targeting does not gives the same robustness as wild type (which also present conformation switch), it would seem natural to try a construct that includes a conformation switch but no membrane targeting. Why was this construct not studied?

On the modelling side.

6) Parameters. For the mathematical model, there are many parameters that have to be set. The authors do a good job of explaining the model, the parameters, and their values (except that *k*_E_ is reported alternately as 5 micron^3^ s^–1^ in the legend to Figure 3 and as 10 micron^3^ s^–1^ in the Parameters in vitro section). However, the reader is left with little insight into how these parameter values were determined. Even if this is somewhat repetitive with previous works, it would be beneficial to add to the current summary of parameter values in the Materials and methods section an additional presentation of all the parameters in the SI with explanations for the sources of the values.

7) The model depends on a limited number of parameters, including minE recruitment rate by minD and minE membrane binding rate, but also on the concentration of minD and minE. The results of the model are shown in 2D phase diagram, without explaining how the pattern formation depends on concentration. It is claimed that dimerisation increases the recruitment rate and membrane targeting decrease minE unbinding rate. While the latter seems reasonable, the former is more questionable. Recruitment could be diffusion-limited, and dimerisation could rather also decrease the unbinding rate. In addition, a dimer of the MTS should have higher recruitment to the membrane just as a dimer of the peptide has higher recruitment to MinD.

These questions could be addressed addressed by extending the modelling to account for the properties of the different construct on the different reaction rates. They could also be addressed by comparing the concentration-dependence of pattern formation with the theoretical predictions.

8) It is found experimentally that the patterned obtained by the different construct have different length scales than the wild type ones. Is it possible to comment on this from the modelling point of view, while still remaining within the realm of linear stability, by discussing how the most unstable wavelength (as in Figure 3—figure supplement 2) varies for the different constructs.

9) The MinE conformational switch. In addition to the features of MinE explicitly addressed in this study, MinE is known to undergo a conformational switch. Indeed, some of the authors recently published a detailed study of the role of this switch (Denk et al., 2018), and concluded that it increases robustness of oscillations to the MinE/MinD ratio. While the dimerizing constructs of MinE used in the experiments may not be conducive to study of the conformational switch, it should certainly be possible to model the separate role of this switch, for example in the absence of the membrane targeting domain.

A final remark:

The predictions for the in vivo behavior of the various MinE constructs are exciting. The current study would have a much greater impact if these experiments were actually performed. While the current study clearly has novelty and general interest, could the effect of the different construct be studied in vivo?

---

## [Author Response]

Essential revisions:On the experimental side.1) The reporting of experimental results rather succinct. One would in particular expect a more thorough analysis of the role of concentration. Figure 2—figure supplement 2. show only two titration curves, while three constructs lead to patterns. To what extent does the wave length of the pattern depend on concentration. These are important issues, since they can also be addressed by the model, and hence reinforce the comparison between the two approaches.

Indeed, we performed a more thorough titration study for the MinE(1-31) and MinE(2-31)-sfGFP mutants (which contain the MTS but no dimerization domain) than for the other mutants/constructs. The role of protein concentrations has now more thoroughly been investigated via modelling (see question 7 of the reviewers’ report). We observe a qualitative agreement between titrations of the MinE(1-31) mutant and phase diagrams obtained theoretically in the “recombination-driven” regime. We also exemplify how a quantitative agreement can be obtained by fitting the kinetic rates of the model.

Our focus in the present study was to demonstrate that protein domains can be combined in a modular fashion, in particular that native domains can be replaced by foreign domains with the same function (here dimerization). For this proof of concept, we tried many different constructs. For instance, we designed and purified all constructs with three different dimerization domains – GCN4, Jun and Fos. Furthermore, we investigated the function in an inducible system – a rapamycin-inducible dimer of the proteins FKBP and FRB. All of these were explored for minimal MinE (13-31) and minimal MinE-MTS (1-31 or 2-31)). Titration of self-organization assays is very labor-intensive. Hence, an in-depth study of the concentration phase diagrams of the various constructs is beyond this proof of principle study. The concentration dependence will be the subject of future works.

Concerning the wavelength of patterns: As mentioned in our answer to question 8, the matter of wavelength selection is an unsolved key problem in the theory of pattern formation far from equilibrium. Furthermore, the new, minimal constructs can sometimes exhibit various (coherent and incoherent) pattern types with vast range of wavelengths within a single assay (see Author response image 1 as an example, taken with MinE(2-31)-msfGFP, and please note that we added a timeseries of this experiment as Figure 2—video 3). A thorough investigation of this rich, multistable phenomenology is beyond the scope of this study. Here, we focused on the system’s ability to form any self-organized pattern in the first place. As the reviewers point out, the detailed investigation of the different pattern types and the question why the minimal constructs exhibit this large variety of pattern phenomena is a promising future direction.

2) When studying the minimal, ATPase activating peptide of MinE, it is found that membrane binding by MinD remains dominant. It is surprising that at high peptide to MinD one does not see a lack of membrane binding by MinD as the peptide has been shown to stimulate the ATPase activity of MinD (paper by Goto in PNAS).

Multiple sources report the stimulation of ATPase activity by the peptide, including ourselves (Glock et al., 2018, where the stimulation was even photo-controlled). However, activity of the peptide seems to become drastically enhanced by adding membrane binding or dimerization, so that these new constructs work at much lower concentrations. As the reviewers rightfully point out, there is a stark contrast between the isolated peptide and the functional minimal MinEs. From previous experiments for the paper mentioned above, we know that the minimal ATPase activating peptide does not become dominant up to very high concentrations (>10 µM).

3) By adding a dimerization or MTS motif the authors find they can recover pattern formation (albeit different from each other). Does the dimerized peptide result in less coherent pattern than the I24N mutant used in a previous study? If so can the authors guess as to what feature of dimerization leads to this difference?

In our study we focused on the ability to form patterns under specific conditions (different minimal MinE constructs), without going into detail on the type of pattern or the coherence of patterns. This is partly due to a gap in theoretical understanding of the properties of highly nonlinear patterns. However, one can safely assume that different pattern types observed in the experiment are the results of differences in system parameters, for example the MinE-MinD association rate. The previously reported I24N construct employed in in vitro studies carries an N-terminal purification tag, which may have an impact on the kinetic parameters (see Glock et al., 2018). Measurements with I24N (or any previously characterized mutant) were done at high magnification and detail. There is an unconscious bias towards showing coherent areas of the assay when working this way. A new study would have to be done on a newly cloned I24N with similar imaging conditions.

4) The text states "but the exact outcome depended heavily on the starting conditions of the assay". What exactly is meant by "starting conditions"? Does this refer to different concentrations, or do the authors observe qualitatively different outcomes even for the same nominal initial conditions including concentrations?

*I*HereBy the term “starting conditions” we refer to the variability between replications of the experiment *with the same protein concentrations*. In a mathematical model simulation this would most closely correspond to different initial conditions and heterogeneities intrinsic to the experimental setup. We shall briefly explain the sources of this variability in the following.

The dimerized constructs with MTS show a very strong membrane affinity, as mentioned before. The established way of starting an in vitro Min assay is to add MinD and ATP, mix well and then add MinE before mixing again. With these constructs, the initial contact area between construct and membrane becomes highly populated with the peptide, and no amount of washing removes this layer. For example, 200 nM of peptide (e.g. MinE(1-31)-GCN4), which would decidedly dominate and inhibit all MinD membrane-attachment, could be added such that a zone is empty while MinD can still bind to areas not reached by the construct. Diffusion on membranes is sufficiently slow to sustain a thus “patterned” state for hours. Therefore, mixing behavior and order of addition can in this case actually create different outcomes at the same average concentrations. We tried to overcome this by pre-mixing MinD, ATP and MinE in a larger volume before addition. However, the random heterogeneities due to imperfect mixing etc. are different each time.

5) Given that minE dimerisation is sufficient to produce pattern of similar wavelength than the wild type, and that dimerisation with membrane targeting does not gives the same robustness as wild type (which also present conformation switch), it would seem natural to try a construct that includes a conformation switch but no membrane targeting. Why was this construct not studied?

The construct suggested by the reviewers is a full-length MinE without MTS, which has been extensively studied by Kretschmer et al., 2017. It was found that this construct leads to very small wavelength oscillations. An in-depth study of the switch by a combination of experiments and theory performed in our groups was recently published (Denk et al., 2018). In particular, it was found that the switch is not essential for pattern formation. Instead, it increases the range of pattern formation to E/D ratios much larger than unity. In the present study, we focused on the minimal requirements for pattern formation and therefore studied only constructs without the switch.

On the modelling side.6) Parameters. For the mathematical model, there are many parameters that have to be set. The authors do a good job of explaining the model, the parameters, and their values (except that k_E_ is reported alternately as 5 micron^3^ s^-1^ in the legend to Figure 3 and as 10 micron^3^ s^-1^ in the Parameters in vitro section). However, the reader is left with little insight into how these parameter values were determined. Even if this is somewhat repetitive with previous works, it would be beneficial to add to the current summary of parameter values in the Materials and methods section an additional presentation of all the parameters in the SI with explanations for the sources of the values.

The only parameter values that have been determined experimentally are the diffusion constants, protein copy numbers (total concentrations) and (to some extend) the nucleotide exchange rate. None of the kinetic rate constants have been measured directly by experiments. Instead they were inferred in (Halatek and Frey, 2012) to reproduce the in vivo phenomenology quantitatively, and to optimize the biological function of the in vivo pole-to-pole oscillation (mid-cell localization). *The* in vitro parameters in (Halatek and Frey, 2018), that we also use here, are based on the inferred in vivo parameter set but adjusted to match the in vitro phenomenology qualitatively and to enable patterns at very large bulk heights. In the revised manuscript we added a table that contains all parameters in both systems (in vivoand in vitro). Finally, the MinE attachment rate (*k*_E_) was indeed stated incorrectly in the legend of Figure 3 of the original manuscript. The correct value is 5 micron^2^ s^-1^. We fixed the legend accordingly.

7) The model depends on a limited number of parameters, including minE recruitment rate by minD and minE membrane binding rate, but also on the concentration of minD and minE. The results of the model are shown in 2D phase diagram, without explaining how the pattern formation depends on concentration. It is claimed that dimerisation increases the recruitment rate and membrane targeting decrease minE unbinding rate. While the latter seems reasonable, the former is more questionable. Recruitment could be diffusion-limited, and dimerisation could rather also decrease the unbinding rate. In addition, a dimer of the MTS should have higher recruitment to the membrane just as a dimer of the peptide has higher recruitment to MinD.These questions could be addressed by extending the modelling to account for the properties of the different construct on the different reaction rates. They could also be addressed by comparing the concentration-dependence of pattern formation with the theoretical predictions.

This comment makes three related points. The first (concentration dependence) and last (comparing the concentration dependence to the titration data from experiments) are closely related, so we will address these two first, and the question on dimerization second.

In the revised manuscript, we added four phase diagrams in the (*n*_E_,*n*_D_)-parameter plane at four different combinations of the MinE recruitment rate (*k*_dE_) and the MinE detachment rate (*k*_e_); see Figure 2—figure supplement 2. For low recruitment and fast detachment, there is no pattern formation at any concentration (which is also what we found in experiments). For the other three (*k*_dE_,*k*_e_) combinations, we that mainly *n*_E_/*n*_D_-ratio determines the regime of pattern formation. This is in qualitatively agreement with the titration experiments for the MinE(1-31) and MinE(2-31)-msfGFP-His constructs, which include the MTS but no dimerization.

Using the concentration-dependence as an experimental constraint is, in principle, an excellent suggestion. To also exemplify how one of the parameter can be fitted to obtain a quantitative agreement, we added a phase diagram in the (*n*_E_/*n*_D_, *k*_e_)-parameter plane at *k*_dE_ = 0 (no MinE recruitment from the cytosol); see Figure 2—figure supplement 2. From this, the value for *k*_e_ which reproduces the experimentally found upper threshold in the *n*_E_/*n*_D_-ratio (roughly 1/20) can be read off. We would like to point out, however, that the parameter-fitting is severely underdetermined since none of the kinetic rates are known in vitro (or in vivo). The available experimental data is unfortunately insufficient to constrain all parameters. (For a different choice of the remaining kinetic rates, the fitted value of *k*_e_ would be different. Such a fit, therefore, is not informative unless all the other parameters are measured or somehow constrained as well.)

In the revised manuscript, we address the question of parameter fitting in the extended discussion of the parameter values (Materials and methods) and in the legend of the new supplementary figures Figure 2—figure supplement 2 and 3.

In Figure 2—figure supplement 2, we also included a phase diagram in the (*n*_E_/*n*_D_, *k*_dE_)-parameter plane for the model without persistent MinE-membrane binding (“*k*_e_ -> infinity”). This phase diagram could be used to compare to future titration experiments with the different dimerization domains (see question 1).

The reviewers also suggest to further extend the model to incorporate more details of the constructs. Such extensions would lead to more unknown kinetic rates that are not experimentally constrained. To learn something from a quantitative comparison between the experimentally determined concentration dependence and various model extensions, these parameters would need to be inferred. As we argued in the previous paragraph, the currently available experimental data are unfortunately not sufficient to constrain the model parameters, already for the model that we use.

Regarding the stronger membrane attachment of a “dimer of the MTS”: Recall that the dimers forms patterns also without any MTS. This shows that dimerization has a significant effect that is independent of the MTS. In our model, we hypothesize that dimerization leads to a stronger recruitment rate to MinD and show that this assumption is consistent with the experimental finding. Also note that our analysis shows that the effect of stronger MinE recruitment dominates over the effect of stronger MinE membrane-binding strength in the presence of a MTS. In other words, the membrane-binding strength of a dimer of the MTS is secondary, to the primary effect of dimerization which facilitates pattern formation independent of the MTS.

Indeed, our findings do not rule out other possibilities what the effect of dimerization might be, other than enhancing recruitment to MinD. More detailed biochemical analysis will be necessary to address this question in future works.

Finally, regarding the question of diffusion limitation: because the diffusion in a vertically extended bulk is explicitly modelled and constrains the flux from the bulk onto the membrane, diffusion limitation of attachment and recruitment processes is inherently part of the model. The dimer has a lower diffusion constant due to its larger size (approximately factor 2), which would lead to a stronger effect of diffusion limitation. This has no significant effect in the model though: we find that reducing the MinE diffusion by a factor of 2 has no significant effect on the pattern forming capabilities of the system.

8) It is found experimentally that the patterned obtained by the different construct have different length scales than the wild type ones. Is it possible to comment on this from the modelling point of view, while still remaining within the realm of linear stability, by discussing how the most unstable wavelength (as in Figure 3—figure supplement 2) varies for the different constructs.

The question of wavelength selection is a major unsolved problem in the theory of pattern formation far from equilibrium. The popular belief that the wavelength of “Turing patterns” is determined by the most unstable mode in the dispersion relation is only true for patterns with small amplitude (i.e. close to homogeneity). This is not the case in the in vitro Min system. The most unstable mode is *not* informative in general far away from the homogeneous steady state, where many modes interact nonlinearly and requires a detailed theoretical analysis beyond linear stability analysis [ref Halatek, 2018, Brauns, 2018]. The general principles underlying wavelength selection remain elusive.

9) The MinE conformational switch. In addition to the features of MinE explicitly addressed in this study, MinE is known to undergo a conformational switch. Indeed, some of the authors recently published a detailed study of the role of this switch (Denk et al., 2018), and concluded that it increases robustness of oscillations to the MinE/MinD ratio. While the dimerizing constructs of MinE used in the experiments may not be conducive to study of the conformational switch, it should certainly be possible to model the separate role of this switch,for example in the absence of the membrane targeting domain.

Modeling and analysis of MinE with the switch has been done in (Denk et al., 2018). Specifically, the switch was modeled both in absence (MinE-mutant L3E) and presence of the MTS (WT MinE); see the section “Relation of MinE mutants to model extensions” in the SI of (Denk et al., 2018) for details. There, the key insight was that the MTS does not play any significant role for pattern robustness. This is in accordance with our findings that in the presence of the dimerization domain, the pathway via dimerization (which we suggest to be recruitment driven) is dominant and the MTS does not matter.

A final remark:The predictions for the in vivo behavior of the various MinE constructs are exciting. The current study would have a much greater impact if these experiments were actually performed. While the current study clearly has novelty and general interest, could the effect of the different construct be studied in vivo?

We are glad to hear that the reviewers find our theoretical predictions for the in vivo behavior exciting. We agree that testing these predictions experimentally is a promising endeavor. Because the Min pole-to-pole oscillations are important for cell division, studying mutants MinE mutants/constructs that exhibit different oscillation patterns (e.g. MinE(1-31) which we predict will exhibit side-to-side oscillations), will be a considerable experimental challenge and goes beyond the scope of the present work.

For instance, we found that the MinE-like constructs we have studied exhibit pattern formation only for MinE-to-MinD concentration ratios below ~1/20. We are currently not aware of well-characterized promoters that would express MinE sufficiently weakly to reach this ratio. Instead of reducing MinE concentration, one could equivalently try to increase MinD concentration. However, high Min protein concentrations have been shown to lead to artifacts like stochastically moving MinD clusters in a previous experimental study [Sliusarenko et al., Molecular Microbiology, 80: 612-627 (2011), https://doi.org/10.1111/j.1365-2958.2011.07579.x].